# QUANTUM ENTANGLEMENT FOR ATTENTION MODELS

## ABSTRACT

Attention mechanisms in deep learning establish relationships between different positions within a sequence, enabling models like Transformers to generate effective outputs by focusing on relevant input segments and their relations. The performance of Transformers is highly dependent on the chosen attention mechanism, with various approaches balancing trade-offs between computational cost, memory efficiency, and generalization ability based on the task.

Quantum machine learning models possess the potential to outperform their classical counterparts in specialized settings. This makes exploring the benefits of quantum resources within classical machine learning models a promising research direction. The role of entanglement in quantum machine learning, whether in fully quantum or as subroutines in classical-quantum hybrid models, remains poorly understood. In this work, we investigate whether quantum entanglement, when used as a resource, can improve the performance of the attention layer in Transformers. We introduce an entanglement-based attention layer within a classical Transformer architecture and numerically showcase scenarios where this hybrid approach proves advantageous. Our experiments on simple standard classification tasks in both vision and NLP domains reveal that the entanglement-based attention layer outperforms classical attention, showing superior generalization on quantum-generated datasets and in settings with limited training data for classical datasets. Additionally, it demonstrates a smaller generalization gap across all tested datasets. Our work contributes towards exploring the power of quantum resources as a subroutine in the classical-quantum hybrid setting to further enhance classical models.

## 1 INTRODUCTION

Machine learning has revolutionized numerous domains by enabling computing systems to learn complex patterns and relationships from vast amounts of data. This capability stems from the utilization of artificial neural networks (ANNs), particularly deep neural networks (DNNs), which are inspired by the structure and function of a human brain. DNNs comprise interconnected layers of artificial neurons, each performing simple computations and transmitting information to subsequent layers. These networks are trained iteratively to adjust the weights and biases associated with them to progressively improve their ability to map input data to desired outputs. Convolutional neural networks (CNNs) have emerged as a particularly successful architecture within deep learning, excelling at tasks that involve analyzing grid-like data, such as images and time series. Their ability to capture local patterns and hierarchical features has contributed significantly to advancements in various fields. However, for tasks involving sequential data, where long-range dependencies and contextual relationships are crucial, CNNs face limitations due to their localized processing nature.

This led to the development of Transformers, a deep-learning model that has become a key part of machine learning. It was first proposed for sequence-to-sequence tasks such as natural language processing (NLP) Vaswani et al. (2017) and later adapted for computer vision tasks Dosovitskiy et al. (2020); Carion et al. (2020), audio processing Dong et al. (2018) and numerous other domains. The architecture has become the backbone of many state-of-the-art NLP models like BERT Devlin et al. (2018), GPT Radford et al. (2018), T5 Raffel et al. (2020), etc.

Transformers depend on the attention mechanism to focus on input segments that might be essential to produce the desired output. The importance of each input is quantified by the weights assigned

to them. These weights indicate the relative importance of each input in the generated output. By incorporating attention, Transformers can selectively attend to the most relevant information, capturing dependencies and relationships within the data. This mechanism is invaluable in tasks such as NLP or computer vision, as it effectively models the relationships between different input segments. The superior performance of these models stems from their ability to learn the correlations characterizing the problem at hand, e.g., the correlations between patches in a typical image and correlations between words in a sentence.

In a seemingly unrelated world, physicists use quantum mechanical wave functions to model complex relations between particles to describe the system accurately. While the underlying physical laws that govern each particle may (or may not) be simple, modeling a collection of particles is complex. The repeated interactions between particles create quantum correlations or entanglement. Hence, the wave function has become an indispensable tool for predicting the properties of quantum mechanical systems made of many interacting particles.

Similarities between a quantum mechanical wave function modeling relationships between quantum particles and a deep neural network modeling the relationship between segments of a high-dimensional input are studied in Levine et al. (2017; 2019). In particular, Levine et al. (2017) explores the structural equivalence between a function modeled by a Convolutional Arithmetic Circuit (ConvAC) and a many-body quantum wave function using the underlying Tensor Network (TN) structure. They make an important observation that the ability of a ConvAC to represent correlations between input regions is related to the min-cut over all edge-cut sets that separate the input nodes when represented using a TN. When the same TN represents a quantum wave function, this quantity is related to a measure of quantum entanglement. Similarly, the expressiveness of a CNN, or equivalently of a many-body wave function, is related to their ability to model the intricate correlation between the inputs Levine et al. (2017). Hence, it is understandable that deep learning models such as CNN and recurrent neural networks (RNN) can efficiently represent highly entangled quantum systems Levine et al. (2019). Again, the TN analysis of these architectures shows an inherent reuse of information in the network. These analogies allow one to borrow well-established insights and tools in quantum mechanics, such as quantum correlation/entanglement measures, to analyze deep neural networks.

Quantum entanglement, which captures correlations beyond classical mechanisms, plays a unique role in this context. Inspired by parallels drawn by Levine et al., we hypothesize that entanglement can be used to model nuanced correlations in classical data, analogous to its role in many-body systems. This idea stems from the observation that quantum systems, with their ability to exhibit entanglement, can capture complex interdependencies that classical models might struggle to represent. By integrating quantum-inspired entanglement measures into classical models, we aim to enhance the ability of these models to capture subtle correlations in data.

In prior work, Cha et al. demonstrated that attention-based quantum tomography captures global entanglement in quantum systems. They speculated that the success of their Attention-based Quantum Tomography (AQT) stems from its ability to model quantum entanglement across the entire quantum system, akin to the way the attention model in natural language processing (NLP) captures correlations among words in a sentence Cha et al. (2021). Furthermore, a Quantum-aware Transformer (QAT) proposed to capture complex relationships between measured frequencies highlights the similarity between highly structured sentences in NLP and the structured measurements in quantum state tomography (QST) Ma et al. (2023). The AQT was shown to outperform other neural network-based models for QST and also demonstrated the ability to accurately reconstruct density matrices of noisy quantum states experimentally realized on IBMQ quantum computers. These advances underscore the potential of quantum-inspired techniques for enhancing classical machine learning tasks.

In contrast to these approaches, our work explores the reverse: we investigate whether quantum entanglement measures, such as entropy, can enhance classical sequence modeling. Rather than focusing solely on the quantum reconstruction of states, we aim to demonstrate that integrating quantum-inspired measures of correlation—particularly entanglement entropy—into the attention mechanism can reveal new insights into classical data modeling. This marks a novel direction that differentiates our work from previous studies, as we integrate quantum entanglement directly into the Transformer model to capture non-classical correlations that traditional attention mechanisms might miss.

Numerous studies have suggested the potential advantages of quantum machine learning models over classical models. For instance, Liu et al. (2021) construct a family of datasets where no classical learner can classify the data with an inverse-polynomial accuracy better than random guessing, while a quantum classifier should, in theory, achieve high accuracy. This result is contingent on the widely believed hardness of the discrete logarithm problem. Similarly, Gyurik & Dunjko (2023) leverage computational hardness assumptions to demonstrate quantum speedups in scenarios involving quantum-generated data, suggesting quantum advantages in a broader range of natural settings, such as condensed matter and high-energy physics. Moreover, Molteni et al. (2024) demonstrate the benefits of using quantum models to learn quantum observables from measured classical data, providing evidence for quantum advantages in certain tasks.

Despite these theoretical findings, practical implementations of quantum subroutines in classical machine learning models have yielded mixed results. Bowles et al. (2024) conducted a comprehensive review of existing quantum machine learning approaches, concluding that classical models consistently outperform quantum models in a direct comparison. Moreover, they found no evidence of improved performance in quantum models relative to classical baselines as problem complexity increases. These observations highlight a significant gap in the quantum machine learning research, specifically regarding the utility and added value of quantum models in real-world tasks.

In this work, to address this gap, we incorporate quantum entanglement into the attention mechanism of a Transformer encoder model. Specifically, we replace the traditional dot product used to compute attention coefficients with the entanglement entropy generated by Parameterized Quantum Circuits (PQC). This integration not only explores a novel use of quantum-inspired methods in deep learning but also provides a new pathway for capturing intricate data correlations through quantum entanglement. This work offers a fresh perspective on how quantum-inspired methodologies can enhance classical machine learning models.

The methodology we follow is as follows:

1. **Quantum embedding:** The query and key vectors are encoded as quantum states using a Quantum Feature Map (QFM).

2. **Entangle quantum states:** The encoded quantum states are entangled using a PQC.

3. **Measure entanglement:** Entanglement entropy between query and key states is computed as attention coefficients.

Thus, the novelty of our work lies in the integration of entanglement entropy into the attention mechanism, marking a significant departure from traditional approaches that rely on dot products or other classical correlation measures. We compare our approach with scaled dot product attention (Vaswani et al., 2017) and test the model on various classical and quantum datasets. The results indicate that i) Entanglement-based attention outperforms classical attention on small-sized datasets. ii) Entanglement-based attention achieves a better generalization gap. For independent verification of the results, we also publish our code online. We provide a detailed description of the methodology, experiments, and results obtained in the subsequent sections.

## 2 RELATED WORK

El Amine Cherrat et al. (2022) proposed a Quantum Vision Transformer capable of handling classification tasks on MNIST datasets. While these models efficiently performed matrix multiplication on quantum states, their performance did not surpass classical counterparts or show any substantial advantage using quantum models.

A recently introduced Transformer model by Khatri et al. (2024), Quixer, uses the Linear Combination of Unitaries (Childs & Wiebe, 2012) to create a superposition of token unitaries and Quantum Singular Value Transform (Gilyén et al., 2019) to further apply a non-linear transformation to this superposition. The model was tested on the Penn Treebank dataset, and the results indicate that its performance is competitive with an equivalent classical baseline. Similarly, the SASQuaTCh architecture (Evans et al., 2024) implements self-attention in a fully quantum setting using Quantum Fourier Transform but lacks comparative analysis.

The Quantum Self-Attention Neural Network (QSANN) introduced by Li et al. (2022) uses a Gaussian projected quantum self-attention mechanism. It outperformed the existing best QNLP model (Lorenz et al., 2023) in text classification tasks. We compare the proposed model with this approach.

Some proposed quantum Transformer models are more theoretical and have limited comparative analysis with classical attention layers. For example, GQHAN: A Grover-inspired Quantum Hard Attention Network by Zhao et al. (2024), and Quantum Algorithm for Attention Computation by Gao et al. (2023), which incorporate Grover's algorithm into the attention mechanism do not show practical analysis. Some works have also designed quantum circuits that implement adapted versions of the Transformer's core components and generative pre-training phases (Liao & Ferrie, 2024; Guo et al., 2024).

[Update: (Shi et al., 2023) propose a method for computing the dot product between query and key vectors by mapping them into quantum states. They evaluate their approach on the MC and RP datasets. (Shi et al., 2022) introduce the Quantum Self-Attention Network (QSAN), where the Quantum Self-Attention Mechanism (QSAM) is implemented using Quantum Logic Similarity (QLS) and a Quantum Bit Self-Attention Score Matrix (QBSASM). They evaluate their work on a binary classification task using the MNIST dataset, which is significantly simpler than the tasks addressed in this study. (Di Sipio et al., 2022) explore the development of a quantum transformer model by replacing the linear layers used to generate query, key, and value vectors with Parameterized Quantum Circuits (PQCs). We remain cautious about the potential advantages of this approach, as they do not provide empirical evaluations on any datasets.]

In contrast to previous studies, we propose an attention mechanism that utilizes quantum entanglement to capture the relationship between query and key vectors. To our knowledge, this is the first work that showcases measures of entanglement in classical machine learning models and also shows specific scenarios where entanglement-based attention outperforms classical attention models.

## 3 ATTENTION MECHANISM IN TRANSFORMERS

Transformers typically have an encoder-decoder structure using stacked attention and fully connected layers along with layer norms and residual connections Vaswani et al. (2017). The attention layer is responsible for relating different parts of a sequence to compute its representation. In the following we describe the simple process of a self attention layer with single attention head. The output of the attention layer is computed by first creating three vectors: query, key, and value vectors from each input or hidden activations and computing the output as follows:

$$Q = W_q Z^\top, K = W_k Z^\top, V = W_v Z^\top \in \mathbb{R}^{d \times N}, \tag{1}$$

$$A = QK^\top \in \mathbb{R}^{N \times N}, \tag{2}$$

$$\text{Attention}(Z) = \text{softmax}(A/\sqrt{d_h})V^\top \in \mathbb{R}^{N \times d}. \tag{3}$$

where $Z \in \mathbb{R}^{N \times d}$ is the input matrix to the attention layer representing $N$ tokens of dimension $d$. $W_q, W_k$, and $W_k \in \mathbb{R}^{d \times d}$ are the query, key, and value matrices of learnable parameters. Note that we do not apply output projection $W_o$ as we only consider one attention head. The attention coefficient matrix $A$ represents the dot product of all query and key vector pairs. The dot product here acts as a measure of similarity between key and query vectors. Our target is to replace this with a quantum-based measurement.

## 4 ENTANGLEMENT-BASED ATTENTION

We propose entanglement-based attention to test the potential of quantum entanglement to capture relationships within datasets, analogous to its role in modeling particle interactions in quantum systems. We integrate quantum entanglement into the attention mechanism of a Transformer. The key, query, and value vectors are computed using different feed-forward layers as in a classical attention layer. The steps involved further are described below.

## 4.1 QUANTUM EMBEDDING

Quantum computers inherently represent data in Hilbert space. Quantum Feature Maps (QFMs) are employed to map classical data into this quantum space. QFMs associate classical data values with physical parameters used to prepare quantum states. Several prominent QFM methods have been proposed by Khan et al. (2024). [Update: In this study, we explore three different encoding techniques for converting query and key vectors into quantum states. i) **Super dense angle encoding:** Here, each qubit is associated with 4 parameterized gates, specifically RX, RY, RX, and RY, with classical features serving as the parameters for these gates. This method requires only one fourth of the number of qubits compared to the number of features. ii) **Dense angle encoding:** In this approach, each qubit is linked to 2 parameterized gates, RX and RY, requiring half the number of qubits as there are features. iii) **IQP encoding:** The Instantaneous Quantum Polynomial-time (IQP) encoding, introduced in (Havlicek et al., 2018), represents $n$ features with $n$ qubits using the diagonal gates of an IQP circuit. This technique provides several potential benefits, such as efficient data representation, exponential data compression, and potential quantum speedup for suitable machine learning applications. The corresponding circuit includes Hadamard gates, RZ gates, and RZZ gates. ]

## 4.2 ENTANGLE QUANTUM STATES

A Parameterized Quantum Circuit (PQC) is applied to entangle the query and key states. A PQC is a quantum circuit with adjustable parameters that can be optimized for specific tasks. The ability of a PQC to generate entanglement, often quantified using the Meyer-Wallach entanglement measure (Meyer & Wallach, 2002), is referred to as its entangling capability. Various studies (Sim et al., 2019; Hubregtsen et al., 2021) have explored the entangling capability of different PQC architectures. Strongly entangling circuits are typically achieved by appending and repeating layers with configurations of two-qubit gates, such as CNOT, CZ, or their parameterized variants.

In our work, we use Controlled-RX gates exclusively in the PQC between the query and key states. This choice emphasizes the circuit's entangling capability over its expressivity. Single-qubit gates are excluded, as they do not contribute to generating entanglement.

[Update: Furthermore, the Quantum Feature Map (QFM) and PQC are applied consecutively in multiple iterations, a technique known as data reuploading, introduced by (Pérez-Salinas et al., 2020). This approach enhances the circuit's expressivity and allows it to capture higher-order correlations that may be missed by single-layer configurations.] Figure **??** illustrates the QFM and PQC architecture.

## 4.3 MEASURE ENTANGLEMENT

We use a measure of entanglement (ME) between the query $|\phi\rangle_{query}$ and key $|\phi\rangle_{key}$ state to compute the attention coefficient matrix $A$. This is described as follows:

$$A = \text{ME}(U_{\text{PQC}}(|\phi\rangle_{query} \otimes |\phi\rangle_{key})) \tag{4}$$

Here, $U_{\text{PQC}}$ represents the unitary applied by the PQC on the query and key quantum embeddings. The attention coefficient matrix $A$ represents the measure of entanglement between all key and query quantum embeddings. We consider the following measures of entanglement.

1. **Entanglement entropy:**
   [Update: The Von Neuman's entanglement entropy of the subsystem $A$ is computed from the density matrix as: $S_A = -\text{Tr}[\rho_A \log(\rho_A)]$. Here, $\rho_A = \text{Tr}_B |\Psi_{AB}\rangle \langle \Psi_{AB}|$ is the reduced density matrix obtained by tracing out the subsystem $B$ from $\Psi_{AB}$, $S_A$ is the Von Neuman's entanglement entropy. The number of measurements required for computing von Neumann entropy using classical shadows Huang & Kueng (2019); Vermersch et al. (2024) is independent of system size and scales quadratically with the precision required. ]

2. **SWAP test:** The SWAP test is a well-known technique for assessing the similarity between two pure n-qubit states $|\phi\rangle_A$ and $|\phi\rangle_B$. Initially, the system is prepared in the state $|\Psi\rangle = |\phi\rangle_A |\phi\rangle_B |0\rangle_C$. A Hadamard gate is then applied to qubit $C$, which is followed by

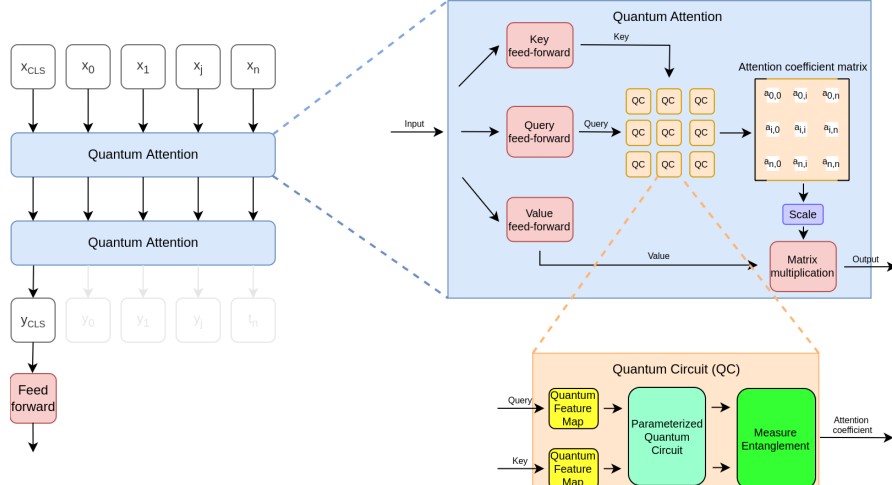

Figure 1: Classical quantum network architecture considered for testing the entanglement-based attention. It is based on the Transformer encoder architecture, consisting of two sequential attention layers and a feed-forward layer. The input is embedded with a class (CLS) token denoted as $x_{\text{CLS}}$, which is used to classify each sample. After the second attention layer, all tokens except the CLS token are discarded, and only the CLS token ($y_{\text{CLS}}$) is passed to the feed-forward layer. For classical attention (used as a baseline), the dot-product between the query and key vector serves as the attention coefficient. For entanglement-based attention, the query and key vector are encoded as a quantum state using a Quantum Feature Map (QFM) and then entangled using a Parameterized Quantum Circuit (PQC). The QFM employs RX and RY gates, while the PQC utilizes CRX gates. The entanglement entropy between the query and key states is subsequently used as the attention coefficient.

a controlled-SWAP gate involving the states $A$ and $B$, with qubit $C$ as the control. The probability of measuring the control qubit $C$ in the state $|1\rangle$ indicates the degree of similarity between $|\phi\rangle_A$ and $|\phi\rangle_B$. We utilize the SWAP test as a base method to evaluate the effectiveness of entanglement measurement.

3. **Using a Modified SWAP Test for Concurrence:** A variation of the SWAP test can be utilized to detect and quantify concurrence, which serves as a measure of entanglement (Foulds et al., 2021). This approach requires two identical copies of the state, denoted $|\phi\rangle_A$ (the original state) and $|\phi\rangle_B$ (the duplicate) for computing entanglement. Moreover, several control qubits, equal in number to those in the test state, must be included, with each initialized to $|0\rangle$. A sequence involving two Hadamard gates and a controlled SWAP gate is applied to each control qubit. Specifically, the SWAP gate acts on $A$ and $B$, swapping the $i^{th}$ qubit of each state only if the $i^{th}$ control qubit is in state $|1\rangle$. The concurrence $C_n$ is then calculated as $C_n = 2\sqrt{P(|\text{even no. of 1s}\rangle_C)}$. [Update: The computational complexity of determining concurrence grows polynomially with the number of qubits involved. This metric provides an assessment of the overall entanglement present within the entire query-key quantum state.]

In our case, the subsystems are the query and key quantum states. A comparison of these entanglement measures was performed to choose the best measure. The experiment results are discussed in Section 5.

## 5 EXPERIMENTS AND RESULTS

To assess the effectiveness of the proposed method, we employed various libraries to implement the hybrid approach. The simulation of quantum circuits was carried out using the TensorCircuit library (Zhang et al., 2023), while the Equinox library (Kidger & Garcia, 2021) was utilized to construct the Transformer architecture. Figure 1 displays the quantum-classical Transformer architecture, which

builds upon the basic Transformer architecture featuring a single attention head. [Update: We apply attention layers in sequence. The combined Query, Key, and Value vectors contribute to a total of $3 * (embed\_dim * embed\_dim + 1)$ trainable parameters. The number of trainable parameters within the PQC in the quantum attention corresponds to half the number of qubits utilized. The final linear layer contains $embed\_dim * n\_classes$ parameters.]

Quantum elements were incorporated into the attention layer, as detailed in the previous section. The query, key, and value vectors were computed from the input using a feed-forward network (without the bias term). These vectors were then mapped to quantum states using a quantum feature map and entangled using a Parameterized Quantum Circuit (PQC). The entanglement entropy between the states was assigned as the attention coefficient. We use only the CLS token output for classification to ensure that the performance of the model primarily depends on the attention layer.

**Evaluation Metrics**   We report three performance metrics for the models: i) train accuracy, ii) test accuracy, and iii) test Nearest Exemplar Accuracy (NEA). We have added the NEA baseline, in order to test the effectiveness of the attention layer in extracting the relevant information for the target classification problem in isolation of the linear classification layer effect and capacity. For that we train the model while omitting the bias term from the classification layer, allowing us to treat the linear classification layer weights as prototypes of the corresponding classes. In the learned embedding space of the CLS token, we can compute another metric of classification accuracy based on the nearest class mean. We compute the mean feature vector of the training samples from each class and then assign to the test sample the label of the closest mean (exemplar) in terms of cosine similarity. We refer to this as Nearest Exemplar Accuracy (NEA). The NEA metric allows us to assess the quality of the extracted CLS token and the learned features in isolation of the optimized classification head.

We report only the interquartile Mean (IQM) of accuracies across ten runs (with different seeds). This was used as an alternative to median and mean as it corresponds to the mean score of the middle 50% of the runs combined across all tasks. This makes it more robust to outliers than mean and a better indicator of overall performance than median (Agarwal et al., 2021).

**Datasets**   We evaluate quantum attention on both classical and quantum datasets. For classical datasets, we use the MC and RP datasets, previously used by Li et al. (2022) for evaluating QNLP models. [Update: MC contains 17 words and 130 sentences (70 train + 30 test) with 3 or 4 words each; RP has 115 words and 105 sentences (74 train + 31 test) with 4 words in each one. The words were converted to vectors using a Word2vec model.]

We also test the model performance on MNIST, FMNIST, and MNIST-1D datasets. MNIST-1D (Greydanus & Kobak, 2024) is a low-dimensional variant of MNIST that emphasizes learning non-linear representations for successful classification. Its small size and complexity make it a suitable dataset for testing quantum models on classical computers. For quantum datasets, we evaluate quantum attention on the Q(E3) dataset proposed by Huang et al. (2021). This is a binary classification dataset (class 0 and 3) generated using the FMNIST and MNIST-1D datasets, employing a Hamiltonian evolution ansatz for classical data embedding and providing Projected Quantum Kernel features as training features.

[Update: Furthermore, all the tokens were represented by a vector of length 12. MC and RP had four tokens each. MNIST and FMNIST images were resized to 12 tokens using bilinear interpolation. The MNIST-1D dataset was reshaped into four tokens. For quantum datasets, 12 qubits were used to generate three tokens.]

**Compared models**   The proposed method was compared with a classical scaled dot product attention-based Transformer. Except for the attention layer, all other layers were identical in both the classical and quantum models. This makes the experiments a fair comparison of these attention models. The hyperparameter settings are detailed in Appendix A. We also compare the models with Quantum Self-Attention Neural Network (QSANN) introduced by Li et al. (2022), which uses a Gaussian projected attention.

In the original QSANN architecture, the mean of the attention layer outputs for all tokens is passed to the feed-forward layer, deviating from the standard classical Transformer architecture. To ensure a fair comparison, we modified QSANN by adding a CLS token to each input and passing only its

| Model | MC Train Acc. | MC Test Acc. | RP Train Acc. | RP Test Acc. |
|---|---|---|---|---|
| Entanglement entropy | 100 | **100** | 100 | **79.26** |
| Concurrence | 80.0 | 73.33 | 75.96 | 70.96 |
| SWAP test | 82.85 | 73.33 | 79.72 | 67.74 |
| QSANN (CLS token) | 58.57 | 56.66 | 67.57 | 54.84 |
| QSANN (original) | 100.00 | 100.00 | 95.35 | 67.74 |
| QSAMb | - | 100.00 | - | 72.58 |
| QSAMo | - | 100.00 | - | 74.19 |

Table 1: **Comparison of various entanglement measures on text classification datasets.** The MC and RP datasets were used to compare the performance of entanglement measures, the SWAP test, QSANN method and [Update: QSAMb, QSAMo models proposed by (Shi et al., 2023).] While the SWAP test is not an entanglement measure, it is commonly used in the literature due to its ability to compute similarity (dot product). We QSANN (CLS token), a modified version of QSANN that uses only the CLS token for classification. The results demonstrate that Entanglement entropy (Von Neumann) outperforms all other measures. Therefore, we adopt entanglement entropy for all experiments in this work.

attention layer output to the feed-forward layer for classification. The experiments conducted to test the quantum attention is described in the following sections:

## 5.1 PERFORMANCE OF ENTANGLEMENT MEASURES ON TEXT CLASSIFICATION DATASETS

The effectiveness of different entanglement measures was tested on the MC and RP datasets. We choose this set due to its small size and its adoption in the studies of Quantum Self-Attention Neural Networks (QSANN). Table 1 summarizes the performance of various entanglement measures and the SWAP test on these datasets. Our results demonstrate that our proposed entanglement entropy outperforms both the original QSANN method and other entanglement measures. Interestingly, the performance of QSANN significantly decreased when only a CLS token is used for classification. The performance of this modified version, denoted as QSANN (CLS token), is also reported in Table 1.

## 5.2 EVALUATION OF ATTENTION MODELS ON QUANTUM GENERATED DATASETS

The attention models were evaluated on the classification tasks using PQK features as proposed by Huang et al. (2021). They show the need for PQK features for classical models (using Multi-Layer Perceptron) to have strong generalization performance. Here, we show the simple classical Transformer encoder cannot generalize well on the PQK feature dataset. However, when it uses entanglement-based attention, our experiments revealed that it succeeds in achieving generalization (Figure 4). This suggests that quantum attention may be particularly effective in leveraging the unique properties of quantum-engineered data.

## 5.3 EVALUATION OF ATTENTION MODELS WITH DIFFERENT DATA SIZES

We investigate the impact of dataset size on model performance on various datasets. Our results revealed a notable trend: the quantum model consistently outperformed the classical model on smaller training sets (Figure 2), while the classical model demonstrated superior performance with larger datasets. This pattern was observed across various datasets, including MC, RP, MNIST, and FM-NIST (Table 2).

We also report the generalization gap on all datasets with varying sizes (Figure 3), defined as the difference between training and test accuracy. The results indicate that quantum attention consistently exhibits a smaller generalization gap across all datasets. This is particularly pronounced for quantum datasets, where the gap is significantly reduced. In some cases, the test accuracy is slightly higher than training accuracy, which might be due to a variance of performance estimation on the different subsets of samples since the test set is 33% of the full dataset. Nevertheless, this suggests that quantum attention may be more effective at preventing overfitting, especially in scenarios with limited training data.

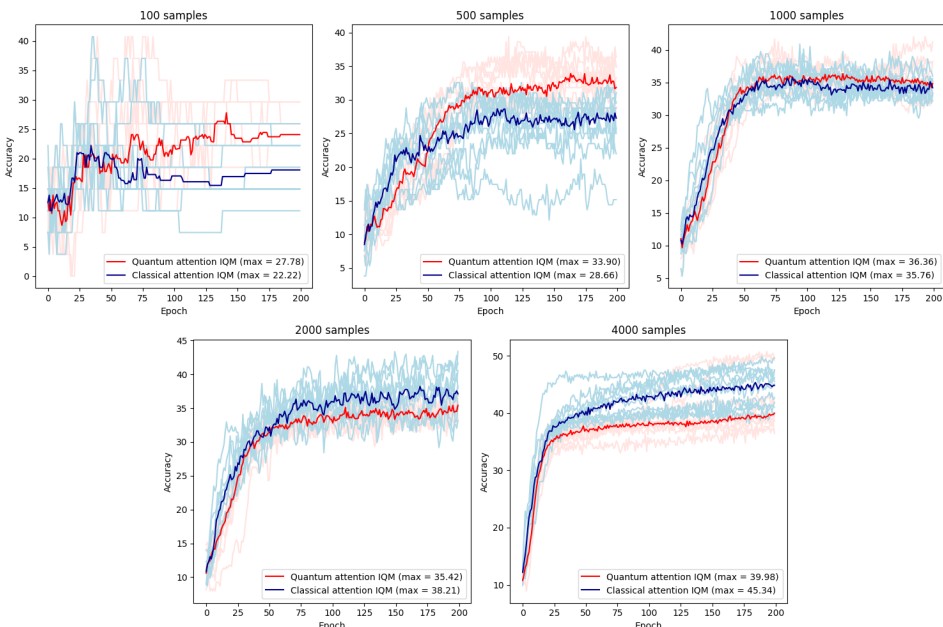

Figure 2: **Test accuracy performance of quantum and classical attention model on varying dataset sizes.** The plot illustrates the Interquartile Mean (IQM) of test accuracy across ten runs of quantum attention on different MNIST-1D dataset sizes. The results demonstrate that the quantum attention model outperforms classical attention on smaller dataset sizes, while the performance of classical attention becomes superior on larger datasets.

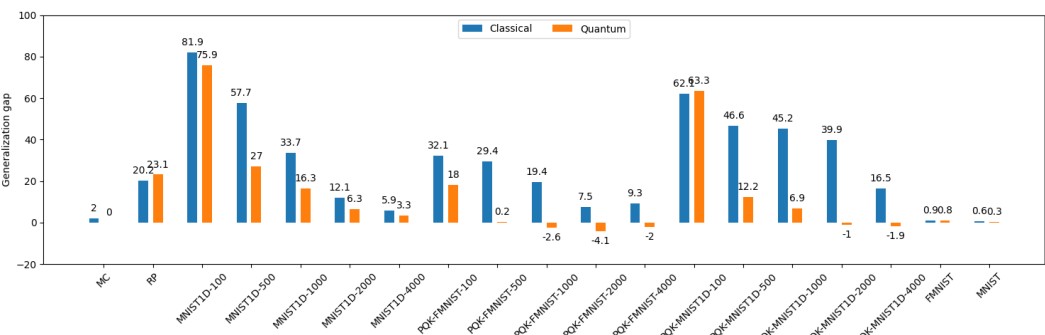

Figure 3: **Generalization gap of quantum and classical attention across datasets.** The generation gap is the difference between train and test accuracy. While quantum attention models may exhibit slightly lower test accuracy on larger datasets, they consistently demonstrate a smaller generalization gap. We show it for all sizes of datasets denoted in *dataset_name–dataset_size* format. A lower generalization gap is better as it suggests that the training accuracy of quantum models is a more reliable indicator of performance on unseen data points, potentially indicating a reduced tendency towards overfitting.

## 6 DISCUSSION

In this work, we introduced quantum attention that uses entanglement entropy to model the similarity between query and key vector. While their performance on large classical datasets did not yield a significant advantage, quantum attention demonstrated superior performance in specific scenarios.

On small datasets, quantum attention exhibited improved generalization capabilities. This was particularly evident on datasets like MC and RP, as well as smaller versions of MNIST and MNIST-1D. This suggests that quantum attention could be a valuable tool for tasks with limited data availability,

| Dataset | # Samples | Train Acc. | | Test Acc. | | Test NEA | |
|---|---|---|---|---|---|---|---|
| | | Classical | Quantum | Classical | Quantum | Classical | Quantum |
| PQK (FMNIST) | 100 | 100 | 99.24 | 75.76 | **84.85** | 79.29 | **84.85** |
| | 500 | 99.32 | 94.14 | 89.66 | **95.01** | 90.16 | **94.66** |
| | 1000 | 97.67 | 92.42 | 85.35 | **93.69** | 85.66 | **93.69** |
| | 2000 | 96.04 | 92.87 | 94.03 | **96.10** | 93.96 | **96.10** |
| | 4000 | 94.91 | 93.38 | 92.65 | **95.35** | 91.67 | **95.35** |
| PQK (MNIST-1D) | 100 | 100 | 100 | 66.23 | **66.67** | **62.88** | 61.21 |
| | 500 | 99.58 | 95.84 | 61.45 | **93.78** | 68.22 | **87.35** |
| | 1000 | 98.25 | 93.24 | 56.16 | **86.41** | 69.29 | **79.88** |
| | 2000 | 96.41 | 93.92 | 91.29 | **94.44** | 84.08 | **94.44** |
| | 4000 | 94.50 | 93.98 | 93.45 | **95.50** | 93.45 | **95.46** |
| MNIST-1D | 100 | 100 | 100 | 22.22 | **27.78** | 24.34 | **27.78** |
| | 500 | 86.29 | 60.07 | 28.66 | **33.90** | 30.15 | **33.96** |
| | 1000 | 67.63 | 52.52 | 35.76 | **36.36** | **38.26** | 37.99 |
| | 2000 | 50.33 | 42.13 | **38.21** | 35.42 | **39.28** | 36.65 |
| | 4000 | 50.52 | 43.63 | **45.34** | 39.98 | **42.90** | 37.86 |
| MNIST | 50 | 100 | 100 | 41.91 | **52.94** | 45.10 | **50.98** |
| | 100 | 100 | 100 | 49.35 | **55.05** | 52.53 | **54.55** |
| | 500 | 100 | 96.42 | **74.42** | 74.06 | **72.63** | 70.10 |
| | 1000 | 99.33 | 86.60 | **78.03** | 75.05 | **76.48** | 69.76 |
| MC | 100 | 100 | 100 | **100** | **100** | **100** | **100** |
| RP | 100 | 94.21 | 100 | 66.82 | **79.26** | 69.12 | **78.80** |
| MNIST | 60000 | 93.88 | 85.38 | **93.34** | 84.98 | **88.13** | 75.03 |
| FMNIST | 60000 | 84.60 | 81.62 | **83.76** | 80.92 | **80.50** | 75.06 |

Table 2: **Performance of super dense quantum attention models on various datasets**. The table presents the maximum Interquartile Mean (IQM) across ten runs (with different seeds) for each dataset. The IQM, representing the mean of the middle 50% of data, is calculated for each epoch, and the maximum value across all epochs is reported. To compare classical and quantum attention, we consider the IQM of test accuracies. Our results consistently demonstrate that quantum attention outperforms classical attention on smaller classical datasets, which is particularly advantageous for datasets like MC and RP with limited samples. Furthermore, quantum attention performs better on quantum generated datasets, PQK (MNIST-1D) and PQK (FMNIST)

| Dataset | Train Acc. | | Test Acc. | | Test NEA | |
|---|---|---|---|---|---|---|
| | Classical | Quantum | Classical | Quantum | Classical | Quantum |
| MC | 100 | 100 | 100 | 100 | 100 | 100 |
| RP | 96.83 | 100 | 63.51 | 78.82 | 65.16 | 76.67 |
| MNIST1D | 75.40 | 89.20 | 35.20 | 35.40 | 34.20 | 35.40 |
| MNIST | 91.31 | 91.77 | 91.17 | 91.19 | 88.05 | 81.48 |
| PQK (MNIST-1D) | 95.24 | 96.39 | 89.81 | 88.79 | 89.54 | 88.47 |
| PQK (FMNIST) | 96.42 | 97.76 | 90.52 | 90.14 | 90.60 | 90.07 |

Table 3: [Update: **Performance of Dense Quantum Attention Models on Various Datasets:** The table compares the performance of the quantum system when using an improved encoding technique and a larger quantum system. It is evident that the generalization gap in the quantum attention models has decreased compared to the super dense encoding case. Notably, the performance of the quantum attention model, especially on the MNIST dataset, has shown significant improvement.]

such as medical applications. Determining the specific dataset size at which classical attention begins to outperform quantum attention remains an open question. Further analysis in this area could help identify the types of datasets that might benefit most from quantum attention.

When applied to quantum-generated features, quantum attention consistently outperformed classical methods, regardless of dataset size. Furthermore, we also conducted experiments (in Appendix B) on relabelled datasets but with classical features. Huang et al. (2021) show that on this dataset, classical models do not generalize well as they do not have access to PQK features. For this dataset, we observed that quantum models had a significant discrepancy between test NEA and test accuracy. We speculate this might be because the classical feed-forward layer overfits the data while the quantum attention layer provides a good representation.

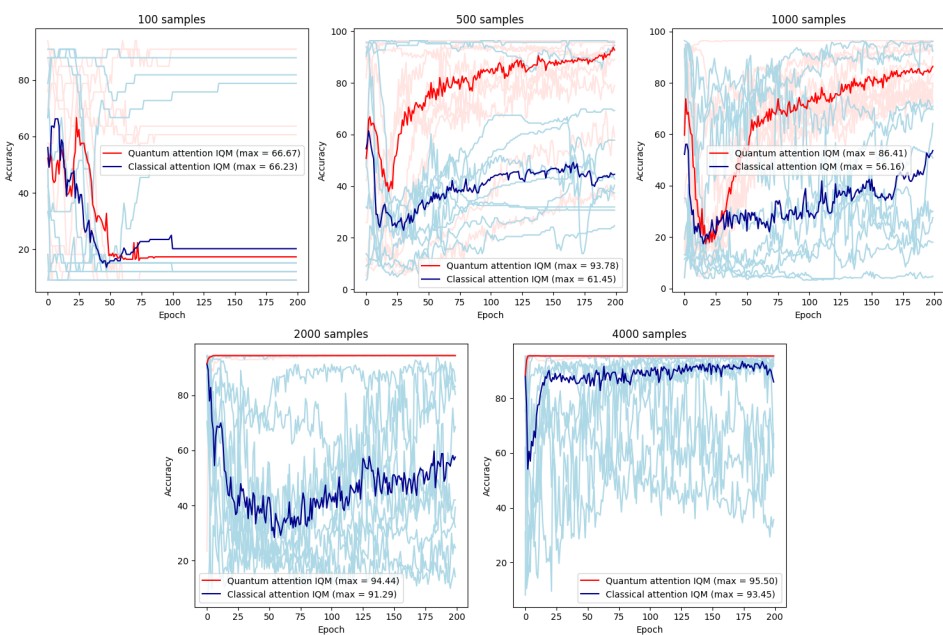

Figure 4: **Test accuracy performance of quantum and classical attention model on a quantum generated dataset.** The dataset used here is the Q(E3) dataset proposed by Huang et al. (2021). This is a binary classification dataset generated using the MNIST-1D dataset (class 0 and 3). The training features are Projected Quantum Kernel features, and labels were generated using the relabelling procedure from Huang et al. (2021). The IQM of test accuracy across ten runs is plotted here. Entanglement-based attention consistently outperforms classical attention for all varying sizes of the dataset and generalizes early on larger datasets, while classical attention exhibits a gradual improvement.

## 7 LIMITATIONS AND FUTURE WORK

To fully realize the potential of quantum attention, future research must address its limitations. These include understanding its behavior under noisy conditions, analyzing its dependence on varying qubit count, identifying suitable application scenarios, investigating the impact of projected quantum features, and exploring the benefits of multiple attention heads. The experiments here were conducted on a classical simulator. Therefore, evaluating their performance on quantum hardware with a larger number of qubits is essential. By addressing these challenges, we can pave the way for practical applications of quantum attention and unlock its full potential in using quantum subroutines in classical machine learning models.

## 8 REPRODUCIBILITY STATEMENT

To ensure the reproducibility of our work, we provide detailed descriptions of the experimental configurations and hyperparameters for the entanglement-based and classical attention model in Section 5 and Appendix A. The source code for all experiments conducted in this manuscript is accessible here: `https://anonymous.4open.science/r/Entanglement-based-attention`.

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

## A  Hyperparameters

[Update:

1. **Embed dimension:** 12 – The length of query and key vectors which are mapped into quantum states.

2. **Optimizer:** Adam – A gradient-based optimization algorithm that adapts the learning rate based on the past gradients to efficiently minimize the loss function during training.

3. **Learning rate:** 1e-2 – The step size used by the optimizer to update the model's weights in response to the gradients, controlling the speed of learning.

4. **Learning rate scheduler:** Cosine – A method for adjusting the learning rate during training using a cosine decay, gradually decreasing it to stabilize training and improve convergence.

5. **Train / test ratio:** 0.66/0.33 – The proportion of the dataset allocated to training (66%) and testing (33%) to evaluate the model's performance.

6. **Data reuploading layers:** 4 – The number of times the classical data is encoded into quantum circuits across successive layers, which helps to enhance the expressiveness of the quantum model.

]

## B  Evaluation of entanglement-based attention on relabelled datasets

We considered the quantum dataset proposed by Huang et al. (2021), where classical PCA dimension-reduced FMNIST features with relabeled classes were used instead of PQK features. They demonstrated that without access to PQK features, classical models struggle to achieve good generalization.

We tested our proposed attention model on this dataset and observed a significant discrepancy between test NEA and test accuracy. This indicates that while the attention layer learns a well-generalizing representation, the subsequent feed-forward layer may be prone to overfitting the training data. We speculate this suggests that incorporating additional quantum components might be necessary to achieve better generalization on this dataset.

We observed quantum attention to outperform classical attention on small dataset sizes.

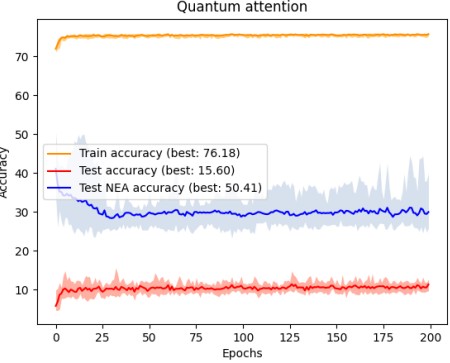

Figure 5: **Evaluation of entanglement-based attention on relabelled dataset.** The performance is on 4000 samples of relabelled samples generated using the FMNIST dataset. There is a clear separation between test and test NEA accuracy.

# C  QUANTUM FEATURE MAPS

Refer to the Figures 6, 7 and 8

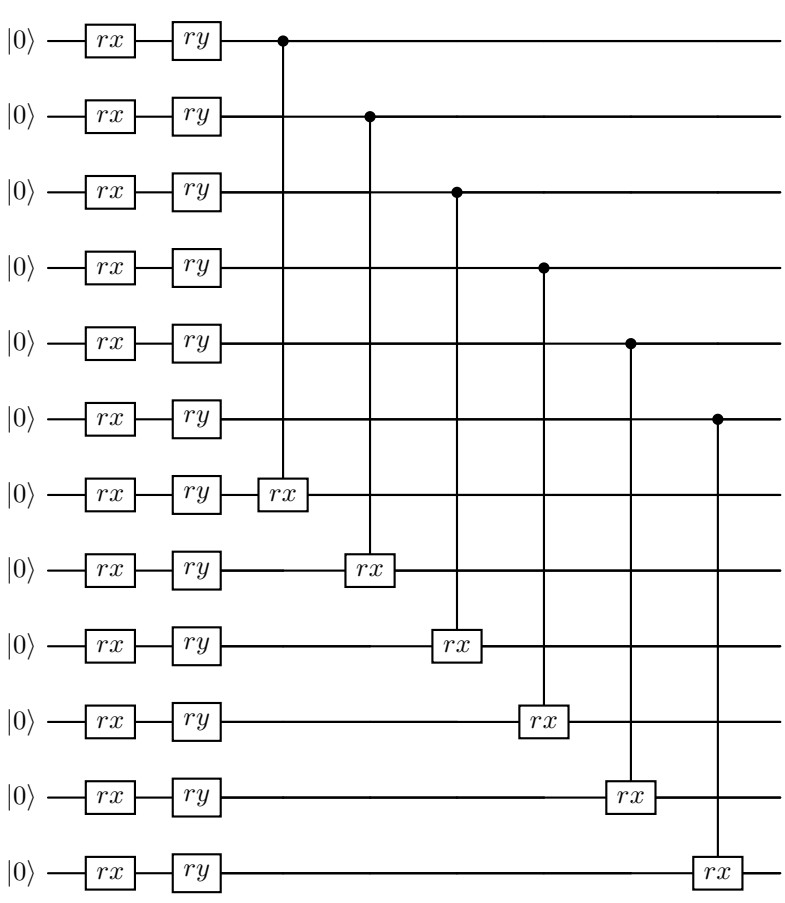

Figure 6: **Dense angle encoding**:Encoding query and key vectors (of length 12) into the parameters of rx and ry gates using 12 qubits.

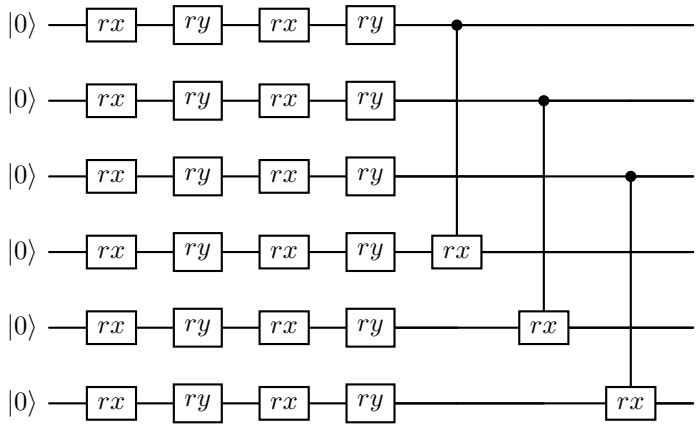

Figure 7: **Super dense angle encoding**: Encoding query and key vectors (of length 12) into the parameters of rx and ry gates using 6 qubits.

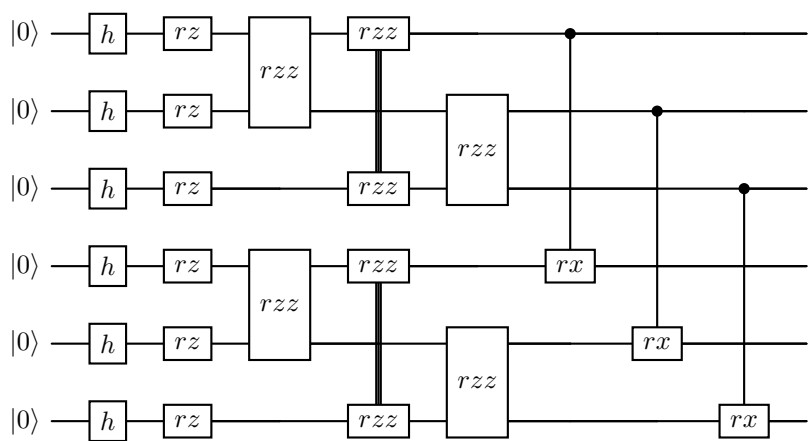

Figure 8: **IQP encoding**: Encoding query and key vectors (of length 3) into the parameters of rz and rzz gates using 6 qubits.

## D  ATTENTION HEATMAPS

In this section, we present the attention coefficients computed by the first layer of both quantum and classical attention mechanisms on the RP dataset. The attention heatmaps are plotted in the Figure 9(b). The attention coefficients provide evidence that the quantum attention layers are capable of focusing on important features in the data. Additionally, these coefficients exhibit distinct patterns for each class, further confirming that the attention layers do not degrade into a simple MLP layer with uniform coefficients.

## E  COMPARISON WITH MLP

In this section, we compare the performance of the quantum attention model with a Multi-Layer Perceptron (MLP). We evaluate the RP and MC datasets using an MLP architecture consisting of three hidden layers and one output layer (Refer Figure 10 and 11). Each hidden layer employs weight matrices of size $48 \times 48$, and the output layer has a weight matrix of size $48 \times 2$, resulting in approximately $7,000$ trainable parameters—substantially more than the quantum attention-based models. Note that, as the RP dataset contains 4 tokens, each of length 12, the dimensions of the weight matrices are $48 \times 48$ and $48 \times 2$.

For the MC dataset, the MLP achieves $100\%$ accuracy on the test set with ease. However, for the RP dataset, the MLP struggles to generalize and exhibits significant overfitting. This experiment highlights that the quantum attention model outperforms the MLP, particularly on the RP dataset. The quantum attention model used for this comparison employs dense encoding.

## F  NOISY SIMULATION

We evaluate the proposed model under noisy conditions, using the MC and RP datasets, which out-perform classical attention and QSANN. Specifically, we employ 1-qubit and 2-qubit depolarizing noise models, as well as thermal relaxation noise, to simulate realistic error conditions for a 12-qubit system under dense encoding. The model is tested with both 2 and 4 attention layers.

Depolarizing noise is modeled as the random application of Pauli gates $(X, Y, Z)$ to qubits, representing the loss of quantum information due to environmental interactions. Thermal relaxation noise simulates the effects of energy relaxation $(T_1)$ and dephasing $(T_2)$.

To ensure realistic simulations, we use the median calibration data from the IBM Kyiv quantum device as of November 19, 2024. The reported values are $T_1 = 277.04 \, \mu s$ and $T_2 = 117.71 \, \mu s$. Depolarizing noise probabilities for single-qubit operations are set as $p_{1x} = p_{1y} = p_{1z} = p_1 =$

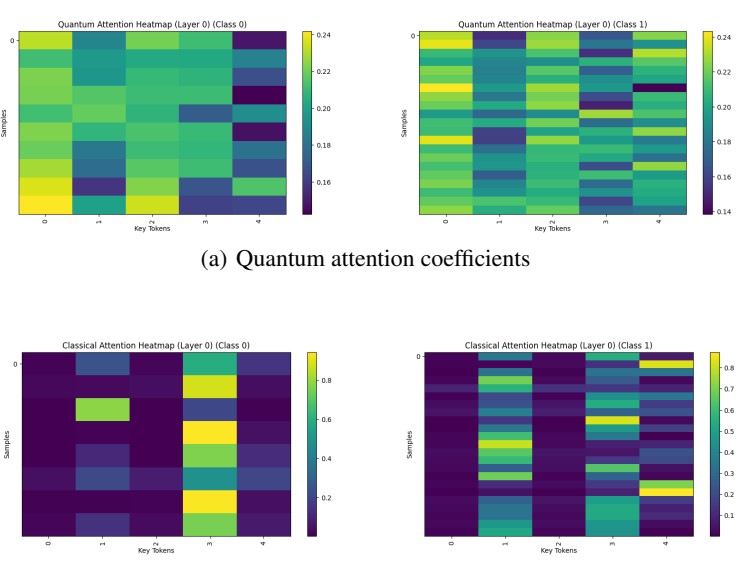

(a) Quantum attention coefficients

(b) Classical attention coefficients

Figure 9: The heatmaps of the attention coefficients for the CLS token, calculated with respect to all other tokens in the RP dataset, are shown here. The coefficients are derived after applying the softmax activation. These heatmaps highlight the attention or importance given by the attention layers to each token while computing the output. Each row corresponds to a sample from a particular class. The plot on the left (right) displays the attention coefficients for all samples belonging to class 0 (class 1) with respect to the CLS token. The samples are grouped according to their predicted class. Both the quantum and classical attention models demonstrate the ability to assign importance to specific tokens, capturing distinct attention patterns for each class. This confirms that both models successfully learn the attention mechanism, varying the level of importance based on class-specific features.

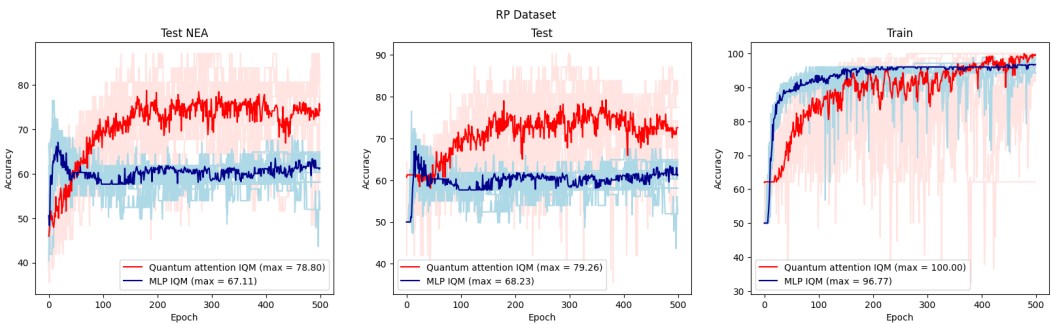

Figure 10: Comparison of quantum attention with MLP on RP dataset.

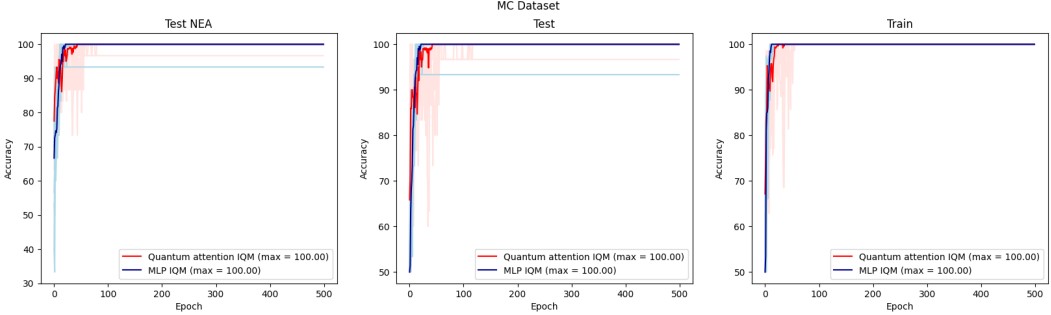

Figure 11: Comparison of quantum attention with MLP on MC dataset.

$2.673 \times 10^{-4}$, while for two-qubit operations, the probabilities are set as $p_{2x} = p_{2y} = p_{2z} = p_2 = 1.224 \times 10^{-2}$.

All simulations are conducted using the TensorCircuit library, leveraging its noise modeling capabilities to evaluate the robustness of the model under these realistic conditions. The results are plotted in Figure 12 and 13.

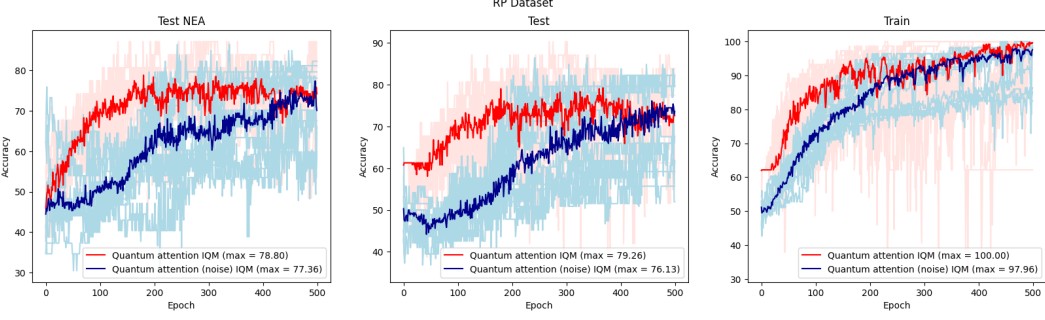

Figure 12: Performance of noisy quantum attention on RP dataset.

# G  PERFORMANCE ON IQP ENCODING BASED QUANTUM ATTENTION

In this section, we evaluate the performance of the quantum attention model using IQP encoding to map classical vectors to quantum states. IQP encoding is considered one of the most "quantum" encoding techniques, potentially offering quantum advantages. However, due to the complexity of the circuit, running simulations with 12 qubits proved challenging on our systems. To address this, we used the MNIST1D dataset, where each token has a length of 3. This allowed us to encode both query and key vectors using 6 qubits. We observed the best training and testing accuracies for

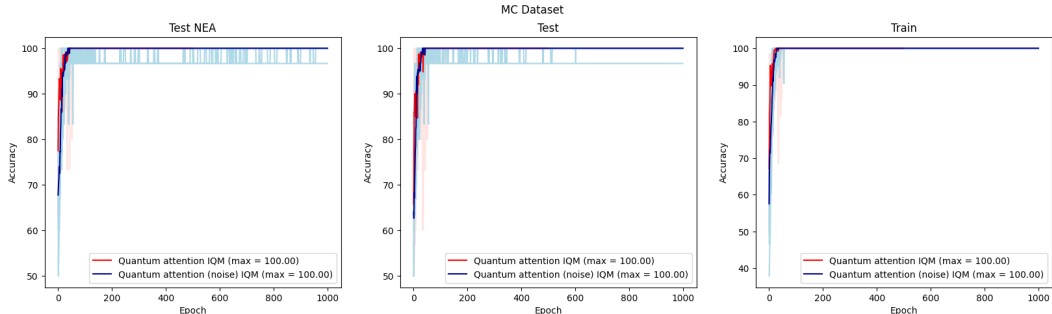

Figure 13: Performance of noisy quantum attention on MC dataset.

quantum and classical attention to be 35.42% and 34.48%, respectively, and 33.85% and 33.18% for classical attention. In this scenario, quantum attention outperforms the classical model.

The simulation times for these circuits were quite high, and in future work, we plan to properly test the model with various architectures and dataset sizes to better understand its scalability.

