# OpenReview forum: "Quantum entanglement for attention models"
_ICLR.cc/2025/Conference — ICLR 2025 Conference Withdrawn Submission_

### Official Review · Reviewer_d6oE · 2024-10-31

**Soundness:** 2
**Presentation:** 2
**Contribution:** 2
**Rating:** 3
**Confidence:** 4

**Summary:**

This paper presents an entanglement-based attention layer integrated into a classical Transformer, where the traditional dot product operation between query and key vector pairs is replaced by a quantum feature map circuit and an entanglement measurement. Leveraging quantum circuits introduces quantum entanglement into the attention mechanism. Numerical experiments indicate that entanglement entropy outperforms other entanglement metrics, and the entanglement-based layer demonstrates advantages over its classical counterpart in classification tasks within vision and NLP domains. For both quantum-generated and classical datasets, the model shows improvements in classification accuracy and a reduced generalization gap.

**Strengths:**

1. The attention mechanism is a cornerstone of modern machine learning, and the potential enhancements offered by quantum computing are compelling.
2. Exploring the synergy between quantum computing capabilities and entanglement is valuable, and this paper provides promising numerical evidence.
3. The quantum circuits are relatively simple and could likely be implemented in near-term quantum computers.

**Weaknesses:**

1. Circuit size: The model proposed in this paper involves a simulated quantum circuit with only 6 qubits; meanwhile, the quantum circuit in Figure 5 is too simple, introducing only local entanglement. Experiments with more qubits (such as 10~20) could significantly improve the soundness of the paper.

2. Motivations: The introduction talks a lot about well-known concepts, but the motivations or insights to replace the dot product with entanglement in the attention mechanism are not sufficiently discussed.

3. Efficiency: My understanding is that the entanglement measurement needs to be performed as many times as the number of attention coefficient matrix elements, which is quite inefficient. The algorithmic/time complexity should be explicitly discussed in this paper.

4. Missing details: There is no specific explanation for the query state and key state; are they row or column vectors of Q and K? Detailed circuit implementations should be provided. Also, the complexities of the different entanglement measurements are not compared in Section 4.3.

5. Concerns about model performance. The numerical results in Figure 2 suggest that the classical model performs better as the sample size increases, potentially diminishing the practical value of this model. I wonder if the small size of the quantum circuit limits performance when using large sample sizes. The training curves are not stable, which could be improved by adjusting hyperparameters. Is there any reason behind the instability of the training curve?

6. Citation mistakes. The introduction refers to 'Systematic benchmarking of existing quantum approaches suggests that entanglement may not play a significant role' but cites no paper. In Section 4.1, QFM methods are reviewed but not proposed by Khan et al. (2024). In Section 4.3, Quantum State Tomography lacks citation, where a typo occurs (FST).

**Questions:**

A few concerning points are listed as follows, and I hope the authors could clarify these before I change my mind in this paper's decision.

1. How about the scalability/complexity of this model, or, what is the scaling with respect to the vector size?

2. Can it show more concrete relations between quantum entanglement and enhancement, to evaluate whether stronger entanglement leads to stronger model performance?

3. If the quantum circuit size becomes larger, will the quantum model keep its advantage on classical datasets as in Figure 2?

4. There have been existing papers differently adapting attention mechanisms, such as Cherrat and Kerenidis (Quantum 2024), Ren-Xin Zhao (IEEE Trans. Pattern Anal. Mach. Intell. 2024), and Khatri (arXiv:2406.04305). What is your advantage or novelty compared to their works?

5. Could you please provide a resource analysis including time complexity, qubit number, or number of measurements...

6. Please clarify several concepts, including "learning rate scheduler", "data reuploading layers" In Appendix A.

7. What is the number of trainable parameters in this model?

---

> ### Author Response · Authors · 2024-11-24
>
> We sincerely thank the reviewer for their detailed feedback and valuable suggestions. Below, we address each concern and question raised. We believe the revisions and clarifications provided significantly strengthen the paper.
>
> ## Response to Weaknesses
>
> 1. **Circuit Size**
>    We agree with the reviewer that the circuit size plays a crucial role in the model's performance. To address this, we have conducted additional experiments with larger quantum circuits by using **Dense Encoding:** and **IQP Encoding**. The details are in Section 4.1, Appendix C and G. We acknowledge that the circuits in Figure 5 primarily introduce local entanglement. We are exploring better entangling circuits as a future direction, as enhanced feature maps and circuit designs could further improve performance.
>
> 2. **Motivations**
>    We appreciate the reviewer’s comment on insufficient discussion of the motivations behind replacing the dot product with entanglement. To address this, we have elaborated on our rationale in the revised introduction:
>
> 3. **Efficiency**
>    The concern regarding the computational cost of entanglement measurements is valid. We have addressed this in the updated manuscript:
>    - **Classical Shadows:** Entanglement entropy can be efficiently approximated using classical shadows.
>    - **Parallelization:** Calculations for query-key pairs can be parallelized, mitigating computational burden.
>    - **Novelty:** This work introduces entanglement entropy to classical ML, laying groundwork for future optimization.
>    A complexity analysis for different entanglement measures is included in Section 4.3.
>
> 4. **Missing Details**
>    We have made the following updates:
>    - **Query and Key States:** Clarified that these are row vectors of query and key matrices.
>    - **Encoding Techniques:** Descriptions of encoding methods (super dense, dense, and IQP) are now in Section 4.1, with corresponding circuit diagrams in Appendix C.
>    - **Complexities:** A comparison of computational complexities is included in Section 4.3.
>
> 5. **Model Performance**
>    We agree that larger quantum circuits and better hyperparameter tuning could improve performance. To address this:
>    - **Larger Circuits:** Expanding to 12 qubits (dense encoding) improved performance, as reported in Table 3 of the revised manuscript.
>    - **Classical vs. Quantum Models:** While classical models perform better with larger datasets, our focus remains on small-data quantum-inspired mechanisms. We observe that larger systems and better encoding lead to overfitting in quantum attention, a limitation we highlight as a future direction.
>
> 6. **Citation Errors**
>    We apologize for the citation errors and have corrected them in the revised manuscript.
>
> ## Response to Questions
>
> - **Concrete Relations Between Entanglement and Model Performance:**
>    Our approach allows the model to learn optimal entanglement via parameterized CRX gates. However, we do not assert a direct relationship between stronger entanglement and better performance. The model’s adaptability to task-specific correlations determines its effectiveness.
>
> - **Performance with Larger Circuits:**
>    Preliminary results using larger systems (12 qubits, dense encoding) suggest improved performance, which we report in Table 3 and Appendix G of the revised paper. We aim to investigate scalability with larger quantum systems in future work.
>
> - **Comparison with Existing Works:**
>    We appreciate the reviewer highlighting related works:
>    - **Ren-Xin Zhao et al.:** Their work introduces quantum kernel-based attention but is limited to MNIST/Fashion-MNIST with a small dataset. Our method shows scalability limitations when implemented on full datasets (all 10 classes), with accuracy dropping to ~10%.
>    - **Cherrat and Kerenidis (Quantum 2024):** They focus on orthogonal layers in quantum transformers and classical preprocessing, which contrasts with our integration of quantum entanglement in the attention mechanism directly.
>    - **Khatri (arXiv:2406.04305):** Their fully quantum transformer represents a significant advancement but is fundamentally different from our hybrid quantum-classical approach focused on classical datasets.
>
> ## Resource Analysis
> The revised manuscript now includes:
>    - **Time Complexity:** A note on the complexity of various entanglement measures.
>    - **Qubit Requirements:** Discussed for each encoding technique.
>
> ## Number of Trainable Parameters
> We confirm that our model has 469 trainable parameters. A breakdown is included in the updated manuscript.
>
> ## Closing Remarks
> We are grateful for the reviewer’s thoughtful feedback, which has led to meaningful improvements in the paper. We hope these clarifications address all concerns. If there are further questions, we would be happy to elaborate.
> We kindly request the reviewer to reconsider their evaluation based on the revisions provided. Thank you for your time and effort in reviewing our work.

---

> > ### Comment · Reviewer_d6oE · 2024-11-25
> >
> > I appreciate the effort that the authors made to revise the paper. However, I cannot be more positive about this work. This decision is derived from the following concerns.
> >
> > 1. The PQC in Fig 6 and 7 can only generate local entanglement between pairs of qubits and has a very shallow circuit depth. From the point of quantum entanglement, it does not show whether such qubit-qubit pairs can generate sufficient entanglement. While from the point of expressivity, it seems to be insufficient as the performance becomes worse for larger sizes.
> >
> > 2. The authors simply dropped some questions in the previous review as "future works", without any illustrations about why it is not included here. Meanwhile, they also neglected some, which is quite confusing.

---

### Official Review · Reviewer_HEkj · 2024-11-03

**Soundness:** 2
**Presentation:** 2
**Contribution:** 2
**Rating:** 6
**Confidence:** 4

**Summary:**

The paper introduces an approach that integrates quantum entanglement into the attention mechanism of Transformer models, proposing an entanglement-based attention layer. By encoding query and key vectors as quantum states, entangling them through a parameterized quantum circuit (PQC), and using entanglement entropy to calculate attention coefficients, the method aims to enhance Transformer performance on specific tasks. Experimental results demonstrate that this quantum-based attention layer outperforms classical attention on smaller classical datasets and quantum-generated datasets, showing a superior generalization gap and reduced tendency to overfit. The work provides valuable insights into leveraging quantum properties within classical machine learning frameworks, especially for data-limited applications, and contributes to the emerging field of quantum-inspired hybrid models. This research lays the groundwork for further exploration of quantum resources as subroutines in machine learning models, particularly Transformers, offering new possibilities for performance improvements in specialized scenarios.

**Strengths:**

This paper presents an innovative approach that integrates quantum entanglement into the Transformer’s attention mechanism, using entanglement entropy to calculate attention coefficients. The method demonstrates improved generalization and reduced overfitting on small classical and quantum-generated datasets, providing a robust evaluation against classical and other quantum attention models. This work contributes to quantum-classical hybrid models, showing potential in data-limited applications and opening avenues for further exploration in quantum-enhanced machine learning.

**Weaknesses:**

1. The paper does not address the impact of noise on the proposed quantum model, which is crucial given the current limitations of noisy intermediate-scale quantum (NISQ) hardware. Quantum systems are inherently sensitive to noise, and without examining how noise affects the model’s performance, it is unclear whether the proposed entanglement-based attention mechanism can be effectively implemented on real hardware. To improve the practical relevance, I recommend adding noise simulations or discussing how hardware noise might affect entanglement performance in attention mechanisms, which would make the work more applicable to real-world quantum devices.

2. Although the authors introduce entanglement entropy for the attention mechanism, the paper lacks a rigorous theoretical foundation to explain why entanglement specifically improves generalization in small-data scenarios. There is little discussion on the advantages of Hilbert space representations (related to quantum feature mapping, QFM) or why quantum entanglement should provide performance benefits over classical models, especially from a quantum information perspective. I recommend that future work include a deeper theoretical exploration of the role of quantum entanglement in attention mechanisms. This could involve discussing Hilbert space properties, parameter efficiency, and the specific benefits of quantum versus classical models, to clarify the approach’s underlying strengths and limitations.

3. The paper does not provide a detailed comparison of parameters between the quantum and classical models, which could help clarify the computational trade-offs of the proposed approach. Including a summary table of model configurations and hyperparameters would enhance transparency, allowing readers to better understand the computational costs associated with each method.

4. The paper only compares its entanglement-based attention mechanism with a simplified Transformer model. It would be helpful to compare against other classical models, such as MLPs, to demonstrate the quantum model’s relative performance more comprehensively.

5. The paper does not reference several recent works that are highly relevant to quantum self-attention and Transformer models. Key papers, such as Shi et al. (2023), Shi et al. (2022), and Di Sipio et al. (2022), explore similar mechanisms and should be cited for completeness. These references would provide additional context and underscore where this work contributes new insights to the existing literature.

**Questions:**

1. Given that current quantum hardware is noise-prone, how do the authors envision the entanglement-based attention mechanism performing in noisy conditions? Are there plans to test this model in simulated noisy environments or on NISQ devices to verify its stability?

2. Although this paper proposes using entanglement entropy for a quantum implementation of the attention mechanism, it lacks an in-depth analysis of the theoretical foundation and effectiveness of this approach. It is recommended that the authors enhance the theoretical exploration of the role of quantum entanglement in the attention mechanism, especially by explaining from a quantum information perspective why it performs exceptionally well on certain tasks. Additionally, a discussion on the theoretical basis and advantages of Hilbert space (related to the Quantum Feature Map, QFM) and the parameter efficiency of quantum models compared to classical models would be beneficial.

3. Can the authors include a table comparing the parameters and architectures of the quantum and classical models to clarify any computational trade-offs? This would help readers understand the efficiency and scalability implications of the proposed approach.

4. Have the authors considered evaluating the entanglement-based attention mechanism against other classical models, such as MLPs, to provide a broader baseline comparison? This could clarify whether the quantum approach offers unique benefits over simpler classical architectures.

5. Several relevant works on quantum self-attention and quantum Transformer models are missing from the current paper. Could the authors consider adding the following references to provide additional context and background on prior work in this area?

Shi, Shangshang, et al. "A natural NISQ model of quantum self-attention mechanism." *arXiv preprint arXiv:2305.15680* (2023).
Shi, Jinjing, et al. "QSAN: A near-term achievable quantum self-attention network." *arXiv preprint arXiv:2207.07563* (2022).
Di Sipio, Riccardo, et al. "The dawn of quantum natural language processing." *ICASSP 2022-2022 IEEE International Conference on Acoustics, Speech and Signal Processing (ICASSP)*. IEEE, 2022.

---

> ### Author Response · Authors · 2024-11-24
>
> # General Response
>
> We sincerely thank the reviewer for their detailed feedback and constructive suggestions, which have significantly helped us improve the paper. Below, we address each point raised, along with changes made to the paper or clarifications to strengthen our contributions.
>
> ## Response to Weakness 1: Noise in Quantum Systems
>
> We appreciate the reviewer's observation about noise in quantum systems. We initially noted this as future work but have since conducted noisy simulations on the RP dataset, where quantum attention showed the most significant performance gain. Specifically, we applied 1-qubit and 2-qubit depolarization noise models, along with thermal relaxation noise, for 12 qubits under dense encoding, with 2 and 4 attention layers. Remarkably, the noisy quantum attention still outperformed the classical attention mechanism. These results have been added in Appendix F of the revised manuscript, along with a discussion of noise models.
>
> ## Response to Weakness 2: Lack of Theoretical Analysis
>
> We acknowledge the need for a deeper theoretical analysis of quantum entanglement in the attention mechanism. Our work was empirical, showing that entanglement entropy can effectively model correlations in quantum-inspired attention. We agree that a rigorous theoretical understanding of its benefits for small-data generalization would be valuable but note that this is beyond the scope of our current study. We have added this point to the discussion as a future direction.
>
> ## Response to Weakness 3: Parameter Comparisons
>
> We appreciate the suggestion to clarify the parameter and computational trade-offs. The primary difference between our quantum and classical models lies in the attention layer, where the dot product similarity is replaced by entanglement entropy via a Parameterized Quantum Circuit (PQC). The PQC introduces additional parameters—specifically half the number of qubits used. We’ve added this response in the revised manuscript.
>
> ## Response to Weakness 4: Comparison with MLPs
>
> Thank you for recommending additional baselines such as MLPs. We tested an MLP with 3 hidden layers and 1 output layer on the RP dataset, where each hidden layer used weight matrices of size (12×4). This resulted in ~7,000 trainable parameters, significantly more than the attention-based models. The MLP still did not generalize well on the RP dataset and showed lower test accuracy compared to both classical and quantum attention mechanisms. These results are included in Appendix E for a more comprehensive comparison of the quantum model’s performance.
>
> ## Response to Weakness 5: Missing References
>
> We have added the references suggested by the reviewer and compared them to our approach:
> - **Shi et al. (2023):** Their quantum state mappings for dot product similarity are evaluated on simpler datasets. Our method outperforms theirs in test accuracy.
> - **Shi et al. (2022):** They propose a Quantum Self-Attention Network but focus on binary classification with MNIST. Our approach generalizes to more complex tasks and uses entanglement entropy.
> - **Di Sipio et al. (2022):** They develop a quantum transformer using PQCs but lack empirical evaluation. Our work fills this gap by demonstrating performance improvements empirically.
>
> These comparisons, along with the added references, strengthen our contribution’s contextualization.
>
> ## Questions Addressed
>
> - **Noise Performance:** Noisy simulations showing model robustness have been added (Weakness 1).
> - **Theoretical Foundation:** Acknowledged as a future direction (Weakness 2).
> - **MLP Comparison:** Conducted and included in the Appendix (Weakness 4).
> - **Relevant References:** Added and discussed (Weakness 5).
>
> ## Closing Remarks
>
> We are grateful for the reviewer’s constructive feedback, which has greatly improved the paper. We hope the additional experiments, comparisons, and references address the concerns raised. If there are any remaining questions or suggestions, we would be delighted to incorporate them.
>
> We kindly ask the reviewer to consider these clarifications and updates when re-evaluating the paper and its contribution. Thank you for your time and thoughtful review.

---

### Official Review · Reviewer_LWay · 2024-11-04

**Soundness:** 3
**Presentation:** 2
**Contribution:** 2
**Rating:** 3
**Confidence:** 5

**Summary:**

This paper investigates the potential of quantum entanglement for attention in Transformers. Authors use the entanglement of quantum states as a co-relation criterion for the Attention layer in the Transformer. The method is evaluated on both language tasks, vision tasks, and quantum datasets. Experiments show the potential of quantum entanglement attention to have better generalization ability.

**Strengths:**

- The paper is easy to follow.
- Introducing quantum in classical computers is meaningful and interesting.

**Weaknesses:**

- Dataset used for experiments is too small. The transformer is a large-scale model that requests large-scale data to learn meaningful features. Besides, quantum entanglement attention shows its outperformance from Figure 2 when the size of the dataset is less than 1000, which is not practical.
- The paper claims that quantum entanglement attention has better generalization ability, which is the difference between train and test accuracy. However, as stated above, the transformer is a large-scale model that requests large-scale data, which means the transformer would easily overfit in small datasets. This has resulted in poor accuracy of transformers on small datasets. For example, in CV tasks, transformers generally require 200-300 epochs on ImageNet to match the accuracy of CNN (which also requires the corresponding number of epochs), while in CIFAR datasets, transformers require 7200 epochs to match the accuracy of CNN, which only requires 200 epochs.
- It's vital to visualize or analyze the attention of quantum and classical. If the elements of quantum attention matrix is all same, the transformer treats all token equally, which means the transformer model is about to degenerate into an MLP which could generalize better than transformer when dataset is small. It's hard to conclude that the benefit is coming from quantum entanglement operation.


- The details of the transformer model should have a description. What's the dimension of the transformer? How many blocks do you stack?
- The details of datasets for experiments should have a more clear description, e.g., MC and RP datasets.
- The details of training should have a description. How many epochs? How do you train the transformer?
- The illustrations in the paper should be improved. The current illustration is confusing and the content is not clear, especially Fig 1.

**Questions:**

See weakness.

---

> ### Author Response · Authors · 2024-11-24
>
> # General Response
>
> We sincerely thank the reviewer for their detailed feedback and constructive suggestions. We appreciate the opportunity to clarify and improve our paper. Below, we address each concern raised and describe the corresponding changes made to the manuscript.
>
> ## Response to Concerns
>
> ### 1. Dataset Size and Transformer Suitability
>
> We respectfully disagree with the notion that Transformers are exclusively suited for large-scale datasets. While Transformers are often employed in large-scale scenarios, numerous studies use simplified architectures or small datasets to analyze specific Transformer components. Our study follows this line, focusing on the applicability of quantum entanglement attention.
>
> - **Performance on RP Dataset**: Our quantum attention model consistently outperforms classical attention-based Transformers, MLPs, and other quantum models on the RP dataset.
> - **Scaling with Larger Systems**: We observed improved performance when scaling the quantum circuit size, indicating that enhancements to the quantum architecture can further benefit the approach.
> - **Novel Contribution**: This work is the first to leverage entanglement entropy in attention mechanisms, demonstrating its potential to enhance performance even on small datasets. While our current experiments focus on smaller data, the methodology can be extended to larger datasets in future work.
>
> We have updated the manuscript to emphasize these points and provide additional context.
>
> ### 2. Visualizing and Analyzing Attention
>
> We appreciate the reviewer’s suggestion to visualize and analyze the attention mechanism. In response:
> - **Attention Heatmaps**: We have added visualizations of the quantum and classical attention matrices in the Appendix. These heatmaps show that the quantum attention mechanism does not treat all tokens equally, validating its functionality and effectiveness.
> - **CLS Token**: The classification in our model relies on the learned CLS token, which effectively aggregates information through attention. This further distinguishes our quantum attention mechanism from MLP behavior.
>
> These results are included in the revised manuscript, with attention heatmaps added to Appendix D and E.
>
> ### 3. Dataset Descriptions
>
> To address the lack of clarity regarding the datasets, we have updated the manuscript to include descriptions of RP and MC datasets and the methodology involved in tokenization. These updates enhance the reproducibility and transparency of our study.
>
> ## Closing Remarks
>
> We are grateful for the reviewer’s constructive feedback, which has led to significant improvements in the manuscript. The updates address concerns about quantum system size, architecture details, and visualization. If further clarification is needed, we would be happy to provide it.
>
> We kindly ask the reviewer to consider these revisions and their impact on the paper’s quality and clarity when re-evaluating the manuscript. Thank you for your time and effort.

---

### Official Review · Reviewer_ANoi · 2024-11-04

**Soundness:** 3
**Presentation:** 3
**Contribution:** 3
**Rating:** 6
**Confidence:** 3

**Summary:**

The paper incorporates quantum entanglement into the attention mechanism of a Transformer encoder by using a measure of entanglement to compute the attention matrix.

**Strengths:**

Novelty:  The paper proposes entanglement based attention and novel methodology for computing attention. Three measures of entanglement(Entanglement entropy, SWAP test and concurrence)  are used for measuring the entanglement between the queries and the value vectors. The proposed method is evaluated in both classical and quantum datasets.  The proposed model was compared with scaled-dot-product attention and another quantum attention method QSANN model for various vision and NLP datasets.
Soundness: The paper is well written. The paper claimed that the quantum entanglement based attention is having a better generalization gap across all datasets. The experimental results of the paper supports this claim. Experiments were conducted extensively on various  NLP and vision  datasets with clear figures and tables.
Significance: With limited number of works done on quantum computing w.r.t Transformers, this work has relevance and future applications
Relation to prior works: The previous related works are discussed comprehensively in this paper.
Reproducibility: The authors have provided the source code of the experiments

**Weaknesses:**

Clarity:  In section 4.3, the authors have defined various methods of entanglement. But how these methods are applied in regard to self attention is not clear. The authors have stated the methods used for computing key, query quantum states and attentions, but exactly how it is done on mathematical terms is not defined. A mathematical expression of  the measures of entanglement for computing the attention, would have made it clearer.  In Figure 5, there is no explanation of how the parameterized quantum circuit is generated.
The  transformer model consists of only  two sequential attention layers. Eventhough the performance of the model with varying data sizes were studied, the performance of the model with varying model sizes have not been studied. Is the model underperforming on larger datasets because of smaller model size?
A qualitative analysis of the behavior of attention maps and the interactions between various positions, if included, would have given a better understanding of the model.

**Questions:**

The model (entanglement entropy) is giving 100% test accuracy on the MC dataset in Table 1. Is there any explanation for this?
In table 1, in the QSANN model, when only CLS token was used the test accuracy dropped from 100 to 56%. What could have been the possible reason?
Why was the comparison only with QSANN model?

---

> ### Author Response · Authors · 2024-11-24
>
> # General Response
>
> We sincerely thank the reviewer for their detailed comments and valuable suggestions. We are pleased that the reviewer engaged deeply with our work and raised critical points that help clarify and improve our presentation. Below, we address each concern raised and provide the corresponding updates or explanations.
>
> ## Clarity on Self-Attention and Entanglement Measures
>
> We appreciate the feedback on the lack of mathematical clarity in Section 4. Classical attention is modeled using softmax on similarities between key and query vectors, computed via the dot product. In our approach, we replace this similarity computation with entanglement measurements derived from quantum states.
>
> To address this concern, we have significantly expanded Section 4 in the revised manuscript. Specifically, we:
> - Provided explicit mathematical definitions for each entanglement measure used to compute attention.
> - Explained the embedding techniques in detail, including how the Quantum Feature Map (QFM) and Parameterized Quantum Circuit (PQC) operate on the quantum states.
> - Added circuit diagrams for both the QFM and PQC to visually illustrate these operations.
> - Included complexity analyses for the various entanglement measurement techniques.
>
> These revisions aim to make the operations and the computation of attention more transparent to the reader.
>
> ## Parameterization and Generation of PQCs
>
> The reviewer noted the lack of explanation for generating the PQC in Figure 5. We clarify that the PQCs in our work are heuristically designed, comprising Controlled-RX (CRX) gates between the key and query states. The model performance and complexity depend on the choice of PQC. A CRX gates-based PQC was chosen to ensure it entangles the query and key state.
>
> ## Study of Model Performance with Varying Model Sizes
>
> We acknowledge the reviewer’s concern about studying model performance with varying model sizes. Due to computational resource limitations, we were unable to test models with more than two attention layers. However, to investigate the effect of a larger quantum system, we conducted additional experiments using a 12-qubit system. The results indicate improved performance with the increased number of qubits, suggesting that larger quantum systems could further enhance the model’s effectiveness. We have included a table in the revised manuscript (Section 5) summarizing these results using dense angle encoding.
>
> ## Qualitative Analysis of Attention Maps
>
> The reviewer’s suggestion to include qualitative analysis of attention maps is highly appreciated. To address this, we analyzed the attention heatmaps produced by our model and observed distinct patterns across different classes. Specifically, we plotted the average attention per class to demonstrate how quantum and classical attention mechanisms differ in their behavior. These visualizations and corresponding discussions have been added to Appendix D of the revised manuscript to provide a better understanding of the model’s attention dynamics.
>
> ## QSANN’s Test Accuracy and CLS Token Usage
>
> We appreciate the reviewer’s insightful questions regarding Table 1. The observed 100% test accuracy on the MC dataset is due to the simplicity of the dataset. When using the QSANN model without the CLS token, the model aggregates information across all tokens for classification. This approach diminishes the role of the attention layer, resulting in behavior more akin to nonlinear layers. By appending a CLS token and restricting classification to this token, as is standard practice in Transformer models, the performance of QSANN dropped significantly to 56%. This suggests that the QSANN’s performance is not primarily attributable to its attention mechanism, underscoring the value of our proposed approach.
>
> ## Comparison with QSANN and Other Models
>
> We focused on comparing our method with QSANN due to the lack of accessible implementations for many existing works. While other models, such as QKSAN, were implemented, they underperformed (e.g., performing no better than random guessing on tasks beyond binary classification). However, following the suggestion of Reviewer 3, we compared our model with Shi et al. (2023), which also utilizes the RP dataset, and found that our approach outperforms their reported results. We have included this comparison in the revised manuscript (Section 5.1).
>
> ## Closing Remarks
>
> We thank the reviewer for their constructive feedback, which has allowed us to significantly enhance the clarity, depth, and presentation of our work. We hope these updates adequately address the concerns raised. We respectfully ask the reviewer to reconsider their score in light of the revisions made and the additional results provided. Please let us know if there are any further questions or areas for improvement.

---

### Note · Authors · 2024-11-29

**Comment:**

Dear Reviewers,

We sincerely appreciate the time and effort you invested in reviewing our paper and providing valuable feedback. Your insights have been instrumental in helping us evaluate and improve our work.

Upon conducting additional experiments, we found that some of our earlier observations regarding the generalization gap and quantum datasets no longer hold true. As a result, we have decided to withdraw the paper to conduct further experiments and refine our findings.

We are committed to revisiting and rewriting the paper with these new insights, ensuring a more accurate contribution to the field.

Thank you once again for your thoughtful reviews and support.

Best regards,
Authors

**Withdrawal Confirmation:**

I have read and agree with the venue's withdrawal policy on behalf of myself and my co-authors.